# Effect of Leaf Litter from *Cistus*
*ladanifer* L. on the Germination and Growth of Accompanying Shrubland Species

**DOI:** 10.3390/plants9050593

**Published:** 2020-05-07

**Authors:** Juan Carlos Alías Gallego, Jonás González Caro, Virginia Hinojal Campos, Natividad Chaves Lobón

**Affiliations:** Department of Plant Biology, Ecology and Earth Sciences, Faculty of Science, University of Extremadura, 06006 Badajoz, Spain; calonge18@gmail.com (J.G.C.); vir_1810_@hotmail.com (V.H.C.); natchalo@unex.es (N.C.L.)

**Keywords:** allelopathy, litter, community composition, *Cistus ladanifer*, shrubland

## Abstract

Most communities with the presence of *Cistus ladanifer* are characterised by the low richness of accompanying species, with *C. ladanifer*, in most cases, exceeding 70% of the coverage of woody species. This fact could be due to the allelopathic activity attributed to compounds present in the leaves of *C. ladanifer*, which may have a negative effect on the germination and growth of woody species that share its habitat. One of the possible ways of incorporating allelopathic compounds to the soil is the degradation of leaf litter. Therefore, the aim of this study was to determine how the presence of leaf litter from *C. ladanifer* affects accompanying species. Under controlled conditions, we analysed the effect of *C. ladanifer* leaf litter on the germination and growth of seedlings of five species that share their habitat with *C. ladanifer* (*Retama sphaerocarpa, Cytisus multiflorus, Lavandula stoechas, Cistus salviifolius,* and *Cistus crispus*). Additionally, the effect of leaf litter on the species itself, *C. ladanifer*, has been studied. The experiments were designed with different concentrations of leaf litter (UL) and leaf litter from which the compounds with allelopathic activity were extracted (WL). The results show that such effect greatly depends on the analysed species, with *L. stoechas* being the most negatively affected species. On the other hand, *C. multiflorus* and *C. salviifolius* were only negatively affected at the stage of seedling growth. The results reveal the involvement of leaf litter in the allelopathic activity attributed to *C. ladanifer* and that its presence has a negative influence on the germination and growth of accompanying woody species. This shows the need to delve into the potential relevance of allelopathy as an interaction that determines the composition, structure and dynamics of a community.

## 1. Introduction

In a global scenario with an increasing climatic variability and an advancing aridity of the Mediterranean basin [1], the natural regeneration capacity of the woody plant species that make up the Mediterranean forest becomes especially important [2]. Shrublands contribute significantly to maintaining the physical integrity of ecosystems susceptible to erosion processes, increase the fertility of soils by providing organic matter [2], reduce soil respiration, thus increasing water availability [3,4], contribute significantly to maintaining biodiversity [5] and act as important carbon reservoirs [6,7].

The composition and structure of a plant community is determined by the net balance between the positive and negative interactions that take place among the different species that constitute it [8]. Facilitation (i.e., positive interactions) is the phenomenon through which one species improves the survival, growth or general state of another species [9,10]. In this sense, previous studies have shown the capacity of shrublands to influence the environment by creating microclimates or markedly different conditions in open spaces, thus allowing the settling of certain species [9,10,11,12].

Similarly, negative interactions determine the final composition of a community. It has been demonstrated that the competition for water and nutrients has a negative effect on the germination and survival of some plant species [13,14]. Another negative interaction is allelopathy [ecological/chemical interaction], where a plant affects the growth and distribution of another species through the production of chemical compounds, with a potential influence on the organization of communities [15]. The allelochemicals involved are mainly compounds derived from secondary metabolism and they are introduced in the environment through leaf washing, root exudate, volatilization and leaf descomposition in the soil [16,17,18]. Understanding this phenomenon can shed light over several important matters: from the ecological points of view, to understand how plants interact through chemicals to determine the structure of plant communities [19,20] and to change the successional processes of vegetation dynamics [21].

The ‘jaral’, a type of Mediterranean shrubland dominated by rockrose (*Cistus ladanifer* L.), constitutes one of the most widespread shrub systems of the western half of the Iberian Peninsula [22], where it prevails in greatly degraded areas with acidic soils [23]. These are secondary shrub communities, mainly derived from the fire destruction of natural sclerophyllous forests, as well as from wood exploitation and the subsequent upper layer soil erosion [24]. This shrub occupies large areas and, in most of this communities, the presence of *C. ladanifer* is above 70% of the coverage of species of these shrublands [25,26,27,28]. The other woody species present, such as *Lavandula stoechas*, *Cistus* sp. and *Halimium umbellatum*, are less abundant [less than 5%] [27,28,29].

One of the greatest problems of these large areas dominated by *C. ladanifer* is the fact that they are normally very poor in species, both herbaceous and woody species [30]. This could be due to the well-known allelopathic activity of *C. ladanifer*, caused by active phytotoxic compounds, which inhibit the development of other plants [31,32,33]. Released into the soil by *C. ladanifer* [34,35], such compounds reduce interspecific competition [31,36], thus decreasing the diversity of accompanying plant species [31]. Allelopathic compounds can persist in the soil for a considerable length of time [37].

In humid regions, allelopathic compounds are released into the soil mainly through the leachate caused by rainfall, whereas the most important mechanisms in more arid climates, such as the Mediterranean climate, are the accumulation and degradation of litter and volatilization. It is worth highlighting that the bioavailability of allelochemicals in the soil depends on transfer and degradation processes [38].

The chemical composition of the leaves of *C. ladanifer* is mainly constituted by flavonoids and terpenes: Apigenin; 4′-O-methylapigenin; 7-O-methylapigenin; 3-O-methylkaempferol; 3,7-di-O-methylkaempferol; 6-acetoxy-7-oxo-8-labden-15-oic acid; 7-oxo-8-labden-15-oic acid; oxocativol or oxocativic acid [39,40]. These compounds follow different routes of incorporation into the soil; diterpenes are transferred to the soil mainly through the leachate of leaves and leaf litter, whereas flavonoids do so through leaf degradation [41].

It has been shown that leaf litter with high concentrations of phenolic compounds hinder the at the stage of seedling growth [42,43]. Therefore, in order to understand how the soil–plant relationships can determine the diversity and composition of a community, it is necessary to know the role of leaf litter as a key element in the incorporation of secondary metabolites into the soil, where they exert their allelopathic activity [44,45].

Thus, considering the relevance of *C. ladanifer* as a dominant species in large areas of the Southwestern Iberian Peninsula, as well as its already demonstrated allelopathic capacity, the aim of the present study was to determine the role of *C. ladanifer* leaf litter in the structure and richness of a community dominated by *C. ladanifer* and its involvement in the allelopathic activity attributed to this species.

To this end, the specific objective of this work was to determine whether *C. ladanifer* leaf litter affects the germination and growth of seedling of several species that are naturally present in typical Mediterranean ecosystems, in order to respond to the following questions: Does the presence of *C. ladanifer* leaf litter limit the germination and settling of seedlings of rival shrub species that compete for resources? Must allelopathy be considered as an element to take into account in the process of natural regeneration of the Mediterranean shrubland?

## 2. Results

### 2.1. Quantification of Compounds with Allelopathic Activity

Table 1 shows the amounts of flavonoids and diterpenes quantified in the untreated leaf litter (UL) and in the washed leaf litter (WL), as well as the percentage of each compound removed after the washing procedure. As can be observed, there is a clear difference in the amount of allelopathic compounds between WL and UL. The compound with the greatest removal percentage was K-3 (98.2%), whereas Ap-4 showed the lowest removal percentage (55.6%).

### 2.2. Effect of Leaf Litter on Germination

The effect of the presence of leaf litter on the germination of the analysed species was quantified through germination percentage (*GP*), germination rate (*GR*) and the time to reach 50% germination (*T*_50_). These parameters were not represented for *C. ladanifer*, since its *GP* exceeded 100%, due to the fact that seeds of this species remained in the leaf litter.

There were two species, *C. multiflorus* and *C. salviifolius*, whose germination was not affected by the presence of *C. ladanifer* leaf litter (Figure 1, Table 2) in any of the three parameters. *R. sphaerocarpa* only showed a delayed *GR* when seeded on 2/3 UL (Figure 1, Table 2).

The two species whose germination was negatively affected were *L. stoechas* and *C. crispus*. The *GP* of *L. stoechas* was lower in the presence of WL, and this negative effect was greater when seeded in UL (Figure 1). In the latter case, there were significant differences with respect to the control, with the two concentrations tested (1/3 and 2/3). In *C. crispus*, when the seeds were seeded in UL, the *GP* was lower with respect to the control. The maximum germination values were reached when the seeds of this species were seeded in WL (showing no significant differences with respect to the control), which could indicate that the contribution of WL to the soil involves a nutrient input that stimulates the germination of *C. crispus*.

Figure 2 shows the germination of two of the studied species—*R. sphaerocarpa*, where the *GP* is not affected (control compared to UL2/3), and *L. stoechas*, where control germination is clearly superior to UL 2/3 treatment.

When quantifying *GR* and *T*_50_, the negative effect of leaf litter was more evident (Table 2). In *L. stoechas*, when the seeds were seeded in UL (1/3), the time required for germination was significantly longer with respect to the control (31.21 days vs. 15.61 days, respectively). There were no significant differences between the treatments of leaf litter in *GR*; on the other hand, *T*_50_ showed significant differences between control and WL 2/3 and UL (13.1 days in the control vs. 24 days in WL 2/3 and 27.1 and 29.4 days in UL 1/3 and 2/3, respectively).

In *C. crispus* (Table 2), the effect of leaf litter on these two parameters was significantly negative when the seeds were seeded in UL, going from 27.4 days required to germinate in the control to 39.7 days in UL 2/3. The shortest *T*_50_ was obtained in the control (23.9 days) and the longest *T*_50_ in UL 1/3 (31.6 days).

### 2.3. Effect of Leaf Litter at the Stage of Seedling Growth

The effect of leaf litter at the stage of seedling growth was quantified by measuring the length of the root and cotyledons (Figure 3 and Figure 4). As can be observed, the parameter that was most affected by the presence of leaf litter was root length, which showed a negative effect in all the analysed species. In *R. sphaerocarpa*, the effect was significantly negative with respect to the control in the four treatments; moreover, there were four species (*L. stoechas*, *C. crispus*, *C. salviifolius* and *C. ladanifer*) in which there was a significant negative effect in UL 2/3 with respect to the rest of the treatments.

It is worth mentioning that the germination of *C. multiflorus* and *C. salviifolius* was not negatively affected; however, root growth was negatively affected by UL 2/3, and even in *C. salviifolius* at UL 1/3.

Regarding cotyledon length (Figure 4), *C. multiflorus* was not negatively affected. In *R. sphaerocarpa*, *L. stoechas* and *C. crispus*, there was a decrease of cotyledon length in the treatments with presence of leaf litter. In *C. salviifolius* and *C. ladanifer*, the decrease of cotyledon length occurred only in UL 2/3.

## 3. Discussion

It has been demonstrated that the presence of shrublands may increase and/or maintain the floristic diversity of plant communities [46]. However, in contrast to the phenomenon of facilitation, it has been observed that, when the environmental conditions become extremely severe, the positive effect of the benefactor may disappear, with a subsequent decrease of diversity [13]. In this sense, allelopathy, as a negative interaction between plants, could be playing an important role in the structure of plant communities, especially under adverse conditions, which increases the production of allelopathic compounds [33,47,48,49].

The exudate of leaves and stems of *C. ladanifer* is constituted by secondary metabolites, mostly phenols and terpenes [30,39,40]. Many of these compounds show phytotoxic activity [30], which demonstrates that they can inhibit the germination and growth of seedlings of herbaceous and shrub species that accompany *C. ladanifer* [31,35]. This implies that such phytotoxic capacity could be the cause of the decrease in the floristic diversity of areas where *C. ladanifer* is settled [32,33]. In this context, the study of the effect of the leaf litter from a species whose allelopathic activity has been reported [30] is a fundamental element to understand the dynamics of plant communities.

The results obtained in this study show that the germination (quantified by *GP*, *GR* and *T*_50_) of *L. stoechas, C. crispus* and *R. sphaerocarpa* are negatively affected by the presence of leaf litter from *C. ladanifer*. On the other hand, no negative effect was observed in *C. multiflorus* or *C. salviifolius*. The results obtained by [35] showed that neither the germination nor the germination rate of *C. salviifolius* was negatively affected when seeded in soils from rockrose thickets.

The present study provides further evidence of the fact that the effect of allelopathic compounds on the germination and growth of other species depends on the target species [32,50], since each of the analysed species was affected differently and in different parameters.

Of the species that were negatively affected, *L. stoechas* was the most affected, with a more pronounced effect on the cultures with UL, suggesting that the presence of allelopathic compounds in the leaf litter are responsible for this behaviour. Therefore, allelopathy would be an interaction to take into account in the growth and development of *L. stoechas*. The influence of allelopathic shrub species on the structure and organization of an ecosystem has been demonstrated in other species, such as *Empetrum nigrum* and *Lonicera japonica* [51,52]. Allelopathic compounds have been found in both species with great inhibitory activity on the germination and growth of seedlings of *Populus tremula*, *Pinus sylvestris* and *Betula pendula*, and these biochemicals have been considered as the causal factor for the species being successful in dominating extensive ecosystems.

Of all the quantified parameters, the most sensitive and negatively affected was root length. When quantifying this parameter, all the analysed species were negatively affected, to a greater or lesser extent, and the test with the UL at 2/3 had a negative effect on all of them. In studies conducted by Chaves et al. [34], it was shown that kaemferol-3-(O) methyl, kaemferol-3,7-di-(O) methyl and apigenin-4′-(O) methyl are involved in the decrease of root length in the herbaceous species included in the study. Moreover, the tests conducted by Alías [33] showed that the root length of *Rumex crispus* is the most affected parameter by the exudates of *C. ladanifer*. This effect on the root affects the opportunity of the plant to settle, which decreases its capacity to make use of the favourable conditions, such as the first rains [33]. Furthermore, the development of the root is important in times when the environmental conditions are adverse, that is, when the competitive relationships are more intense and the facilitating interactions may disappear [13]. Effects on shoot and root growth are equally capable of shaping not just competitiveness for access to resources but also seedling viability. The successful establishment of a species in the Mediterranean region requires a well-developed root system for the efficient capture of resources [53], particularly when water uptake is a limiting factor. The inhibition of the root system development can greatly diminish the performance of more sensitive plants, making them less competitive and, more importantly, less tolerant to drought [35]. With respect to their role in allelopathy and the inhibition of seedling root growth, the activity of flavonoids as regulators of auxin transport and degradation is likely to be of particular importance. Depending on their structure, flavonoids can impact the breakdown of auxin by IAA oxidases and peroxidases [54] and also affect polar auxin transport [55], thereby impacting the root growth of target species.

Furthermore, the present study corroborates the autotoxicity of this species, as was evidenced in studies carried out by Alías et al. [56] and Chaves et al. [57]. This effect is shown in terms of seedling development, since a clear negative effect is observed on root and cotyledon length in the UL treatments at 2/3. This autotoxic behaviour could be involved in the species’ own population control, and would explain the scant self-regeneration within established *C. ladanifer* stands [56]. When a fire affects to *C. ladanifer*, activating the seeds’ germination capacity, only a minority of those seeds end by establishing themselves despite the high percentage of germination. The accumulation of allelopathic substances in the soil from leaf litter of *C. ladanifer* could be related to this fact. Studies carried out by other authors [58,59], where the autotoxicity of species like *Medicago sativa* is demonstrated, defend that ecological advantages of autotoxicity are difficult to interpret, but it encourages geographical distribution of the donor species, serves as an adaptation to induce dormancy and prevents decay of its seeds and propagules. From a practical perspective, the autotoxicity is believed to be a protection mechanism for controlling competition and likely evolved by natural selection.

The effect of leaf litter from different types of species on the growth and establishment of other species has been studied by other authors [60]. It has been observed that the accumulation of leaf litter protects seedlings from the direct impact of rainfall and hail, which are especially common in mountain habitats; leaf litter also protects seedlings from the low temperatures of winter, which constitute an important mortality factor for species with fragile roots, as they cause these to unearth or fracture [61]. Moreover, fallen leaves enrich the soil, which favours the availability of nutrients for seedlings that grow underneath the crowns [13]. All these studies show the positive effects of leaf litter; however, as is shown in the present study, it can also hinder the establishment and growth of other species. These results may be related to the presence of allelopathic compounds (mainly phenols and terpenes), which are released in leaf decomposition, as has been demonstrated by studies conducted by Chaves et al. [41].

The allelopathic activity attributed to *C. ladanifer* not only derives from the tests performed with the compounds obtained from the above-ground part of the plant, as it has been reported that these compounds are present in the soil [34,62], with a varying distribution throughout the year, both qualitatively and quantitatively [34,62]. Demonstrating the presence of these compounds in the soil is essential to evaluate the involvement of allelopathy in the composition of a community, although it is also necessary to demonstrate that the persistence of these phytotoxic compounds is high, since a rapid degradation or transformation could result in a loss of their phytotoxicity [63]. It has been shown that the persistence of the flavonoids detected in soils with presence of *C. ladanifer* is very high, remaining for several months [37]. This high persistence can be due to its continuous incorporation from leaf litter in the degradation process of the latter. The results of our study show the importance of the presence and degradation of leaf litter from *C. ladanifer* in the influence of this species on the germination and growth of seedlings of accompanying shrub species.

It was also observed that some species were negatively affected in the quantified parameters when treated with WL. This could be due to two reasons. It must be taken into account that, although washing the leaf litter greatly reduced the amount of allelopathic compounds in it, as is shown in Table 1, these were not completely removed, thus the cultures with WL contained allelopathic compounds (although in very low amounts). Considering that the effect of allelopathic compounds depends, in many cases, on the joint action of several allelopathic compounds in specific proportions rather than on the separate amount of each [64,65], the amount and proportion maintained in WL could have been enough to inhibit the germination and growth of the analysed species. On the other hand, it could also be due to the action of leaf litter as a mechanic barrier to the growth of seedlings, as has been demonstrated in studies carried out by Cleavitt et al. [66]. This could explain the observed inhibition of the measured parameters in some of the treatments with WL.

In general terms, considering the effect of leaf litter from *C. ladanifer* without differentiating between UL and WL, and quantifying this effect on the germination percentage and root length as the mean of all the analysed species (Figure 5), it is clearly observed that leaf litter from *C. ladanifer* has a negative influence on these two parameters, and, consequently, the floristic composition and structure of the communities where this species is present can be negatively affected. Furthermore, considering the effect of leaf litter from *C. ladanifer* differentiating between UL and WL (Figure 6), it is observed that the negative effect is more pronounced in the UL.

The results presented in this study show the involvement of leaf litter in the allelopathic activity that *C. ladanifer* exerts in the communities in which this species is present, as well as the need to delve into the potential relevance of allelopathy as an interaction that determines the composition, structure and dynamics of a community.

## 4. Materials and Methods

### 4.1. Collection of Leaf Litter and Preparation of Substrates

*C. ladanifer* leaf litter was collected directly from the soil in a rockrose thicket (‘jaral’) located in the municipality of Perales del Puerto, Cáceres (Spain) 40°7′16″ N–6°39′28″ W.

The collected material was part of a layer of leaf litter that covers, and is part of, the substrate where the rockrose thicket is settled, which had fallen at the end of summer and beginning of autumn.

This leaf litter was taken to the laboratory in hermetic and dark bags. Then, it was cleared of stones and remains of herbaceous vegetation.

To verify whether the compounds present in the leaf litter are the ones responsible for the allelopathic activity attributed to *C. ladanifer* [30], the tests were conducted with leaf litter from which most of the content of allelopathic compounds was removed (WL) and untreated leaf litter (UL). To this end, the collected leaf litter was divided into two equal parts, and the following procedure was applied to one of them:

The leaf litter was submerged in methanol (0.5 g/mL) and shaken for 10 min in a JP SELECTA S.A. shaker, to favour the extraction of the allelopathic compounds. Then, the methanol was removed, and the leaf litter was submerged again in a methanol:water solution (1:1) (0.5 g/mL) and shaken for 12 h. Lastly, the leaf litter was washed several times to remove the remaining methanol, after which it was left to dry at room temperature.

The substrates used in the germination tests, which define each treatment, were prepared by mixing and homogenising vermiculite with leaf litter in the following amounts:Vermiculite (control): substrate composed only of vermiculite (2 dm^3^).Untreated Leaf Litter ⅓ (UL 1/3): 1.33 dm^3^ vermiculite and 0.66 dm^3^ untreated leaf litter.Untreated Leaf Litter ⅔ (UL 2/3): 0.66 dm^3^ vermiculite and 1.33 dm^3^ untreated leaf litter.Washed Leaf litter ⅓ (WL 1/3): 1.33 dm^3^ vermiculite and 0.66 dm^3^ washed leaf litter.Washed Leaf litter ⅔ (WL 2/3): 0.66 dm^3^ vermiculite and 1.33 dm^3^ washed leaf litter.

### 4.2. Target Species

The shrubland species selected as target species were: *Retama sphaerocarpa, Cytisus multiflorus, Lavandula stoechas, Cistus salviifolius, Cistus crispus* and *Cistus ladanifer.* These species of the Mediterranean shrubland were selected for sharing their habitat with *C. ladanifer* [23,24,25,26,27,28,29]. The seeds were purchased from Semillas Silvestres S.L. (http://www.semillassilvestres.com) and, except for *L. stoechas*, they showed dormancy [67,68,69], which is defined as the state in which the seeds do not germinate despite having the normal conditions to do it, due to internal physical and/or physiological mechanisms of the seed [70]. Therefore, in order to obtain an optimal germination, these seeds had to be subjected to certain conditions that unlock such mechanisms. Thus, the seeds of the selected species in this study were subjected to the following treatments:*C. ladanifer, C. crispus* and *C. salviifolius*: heat treatment (100 °C) for 5 min [67,68].*R. sphaerocarpa* and *C. multiflorus*: water scalding at 80 °C for 30 s followed by cooling [69].*L. stoechas*: no pregerminative treatment.

### 4.3. Experiment Design

Each species was subjected to five different treatments (Control, UL1/3, UL2/3, WL1/3, WL2/3; 4 replicates per treatment). In PVC trays of 3 dm^3^ (35 × 23 × 7.5 cm), 2 dm^3^ of the substrate was placed, where 25 seeds of each species were seeded (100 seeds per treatment), except for *R. sphaerocarpa*, of which 15 seeds were seeded (60 seeds per treatment) due to its larger size. The germination tests were conducted in a greenhouse under the following conditions: 12 h of illumination (20–25 °C) and 12 h of darkness (10–15 °C). The trays were watered with distilled water until saturation every two days. The number of germinated seeds was recorded daily. After 50 days, the seedlings were extracted and the length of their roots and stem was recorded.

With these data, the following parameters were obtained:-Germination percentage (*GP*): The final germination percentage was calculated using the following formula:*GP* = number of germinated seeds/number of seeded seeds.-Germination rate (*GR*): The germination rate is an arithmetic mean that indicates the days required for germination [71]. It was calculated using the formula cited by [72]:$$GR=\frac{N1G1+N2G2+\ldots \ldots +NnGn}{G1+G2+\ldots \ldots +Gn}=\frac{\sum_{i=1}^{n}NiGi}{\sum_{i=1}^{n}Gi}$$
where *GR* is the germination rate; *N*_1_, *N*_2_, …, *Nn* represent the number of days from the beginning of the germination test; and *G*_1_, *G*_2_, …, *Gn* represent the number of germinated seeds at day *n*.-The time to reach 50% germination (*T*_50_). It was calculated according to the following formula [73]:$$T50=\frac{ti+\left(\frac{N}{2}-ni\right)\times \left(tj-ti\right)}{nj-ni}$$
where *N* is the final number of emergence and *ni*, *nj* cumulative number of seeds germinated by adjacent counts at times ti and tj, respectively when *ni* < *N/2* < *nj*.-Root and stem length: Immediately after removing the seedlings from the culture media, the length of the roots and above-ground part of each individual was measured, with millimetric precision.

### 4.4. Quantification of Allelopathic Compounds

The allelopathic compounds (flavonoids and diterpenes) present in the washed leaf litter (WL) and untreated leaf litter (UL) were quantified by high-performance liquid chromatography (HPLC). To this end, 2 g of sample (three replicates of each) was submerged in 2 mL of chloroform for 10 s [39]. Once the chloroform evaporated, the sample was submerged in 2 mL of methanol, filtered with a 13 mm–0.45-µm filter paper, and then frozen at −18 °C for 12 h. This procedure ensures the precipitation of waxes, which were removed by centrifugation at 10,000 rpm. Then, the diterpenes and flavonoids were quantified by HPLC (Waters, 515 HPLC pump, 717 autosampler injector, 996 photodiode array detector). Next, 25 µL of sample was injected in a reverse phase column (Spherisorb 5µ C-18 4.6 × 250 mm). The mobile phase used was water:methanol:tetrahydrofuran (56:16:28), at flow rate of 0.75 mL/min. With these conditions, the resolution of the chromatogram was optimal for the quantification of the flavonoids and diterpenes present in the exudate of *C. ladanifer* [39,40].

### 4.5. Statistical Analysis

In every case, the normality and homoscedasticity of the variables were explored using the Shapiro–Wilk test and Levene’s test, respectively.

To assess the effect of the treatment on the different variables, the one-way ANOVA test or the Kruskal–Wallis non-parametric test was used accordingly. The subsequent analyses for the significant cases were conducted using Tukey’s HSD test or the Mann–Whitney non-parametric test. All statistical analyses were carried out using the SPSS statistical software.

## Figures and Tables

**Figure 1 plants-09-00593-f001:**
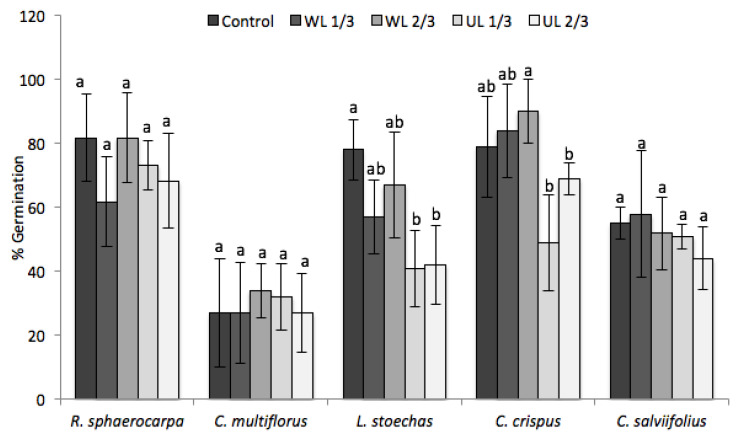
Effect of leaf litter on the germination percentage of the analysed species in the five treatments (C: vermiculite control; WL, washed leaf litter; UL: untreated leaf litter; 1/3: 1.33 dm^3^ of vermiculite and 0.66 dm^3^ of leaf litter; 2/3: 0.66 dm^3^ of vermiculite and 1.33 dm^3^ of leaf litter). a, b: same letter means no significant difference between treatments (*p* > 0.05).

**Figure 2 plants-09-00593-f002:**
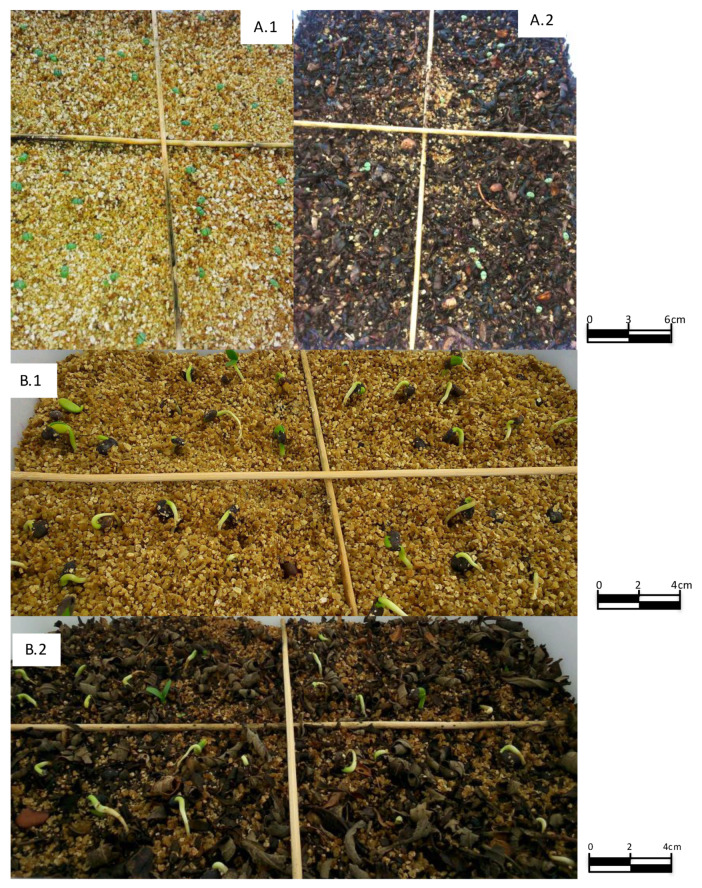
Image of the germination of *L. stoechas* seeds sown in the control (**A.1**), and sown in UL2/3 (**A.2**). This image has been taken 15 days after the seeds were sown. Image of the germination of the *R. sphaerocarpa* seeds sown in the control **(B.1**), and sown in UL2/3 (**B.2**). This image has been taken 10 days after the seeds were sown. Control: vermiculite; UL2/3: untreated leaf litter, 0.66 dm^3^ of vermiculite and 1.33 dm^3^ of leaf litter.

**Figure 3 plants-09-00593-f003:**
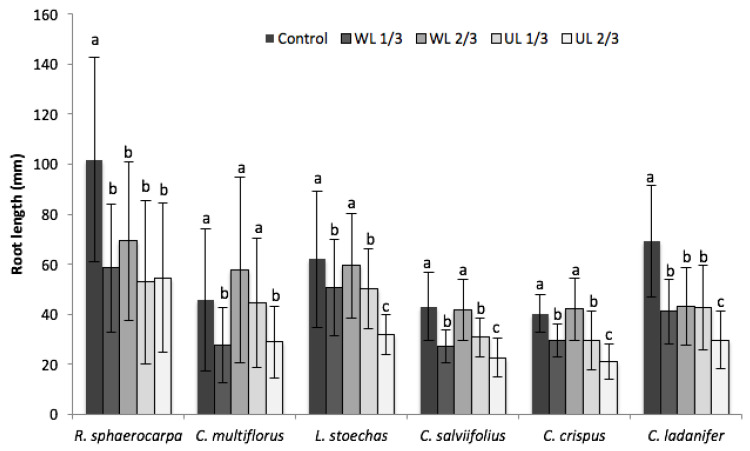
Effect of leaf litter on the root length of the analysed species in the five treatments (C: vermiculite control; WL, washed leaf litter; UL: untreated leaf litter; 1/3: 1.33 dm^3^ of vermiculite and 0.66 dm^3^ of leaf litter; 2/3: 0.66 dm^3^ of vermiculite and 1.33 dm^3^ of leaf litter). a, b, c: same letter means no significant difference between treatments (*p* > 0.05).

**Figure 4 plants-09-00593-f004:**
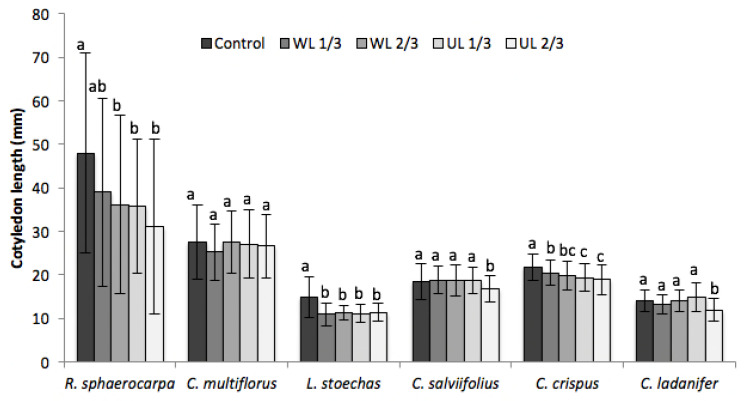
Effect of leaf litter on the cotyledon length of the analysed species in the five treatments (C: vermiculite control; WL, washed leaf litter; UL: untreated leaf litter; 1/3: 1.33 dm^3^ of vermiculite and 0.66 dm^3^ of leaf litter; 2/3: 0.66 dm^3^ of vermiculite and 1.33 dm^3^ of leaf litter). a, b, c: same letter means no significant difference between treatments (*p* > 0.05).

**Figure 5 plants-09-00593-f005:**
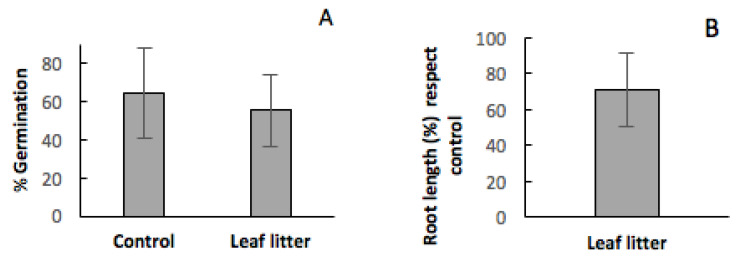
Effect of the presence of leaf litter from *C. ladanifer* on germination percentage (panel (**A**)) and root length with respect to the control (panel (**B**)). These parameters were quantified as the mean of all the analysed species.

**Figure 6 plants-09-00593-f006:**
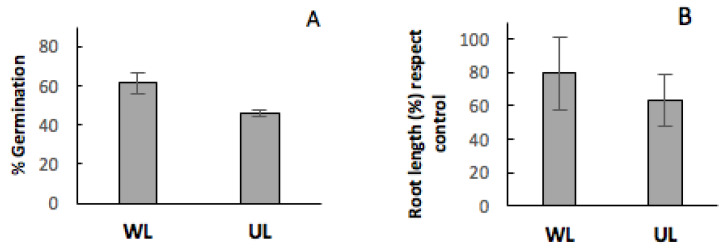
Effect of the presence washed leaf litter (WL) and untreated leaf litter (UL) from *C. ladanifer* on germination percentage (panel (**A**)) and root length with respect to the control (panel (**B**)). These parameters were quantified as the mean of all the analysed species.

**Table 1 plants-09-00593-t001:** Amount of flavonoids and diterpenes (μg/g dry weight) present in untreated leaf litter (UL) and washed leaf litter (WL). Values are the mean of three replicates ± standard deviation.

Compound	UL (μg/g dw)	WL (μg/g dw)	Quantity Removed (%)
Ap	6.0 ± 0.4	1.2 ± 0.1	80.6 ± 2.7
Ap-4	57.8 ± 3.2	25.6 ± 0.1	55.6 ± 2.6
Ap-7	385.5 ± 10.4	44.1 ± 1.4	88.5 ± 0.2
K-3	85.8 ± 17.1	1.5 ± 0.1	98.2 ± 0.4
K-3,4	86.1 ± 10.6	12.7 ± 0.2	85.1 ± 2.0
K-3,7	2564.7 ± 114.7	126.6 ± 15.7	91.5 ± 0.6
D1	25.8 ± 0.8	2.5 ± 0.1	90.1 ± 0.7
D2	2.2 ± 0.1	0.3 ± 0.0	88.1 ± 1.0
D3	3.9 ± 0.9	0.4 ± 0.0	88.8 ± 2.8

Ap: apigenin; Ap-4: 4′-O-methylapigenin; Ap-7: 7-O-methylapigenin; K-3: 3-O-methylkaempferol; K-3,4: 3,4′-di-O-methylkaempferol; K-3,7: 3,7-di-O-methylkaempferol; D1: 6β-acetoxy-7-oxo-8-labden-15-oic acid; D2: 7-oxo-8-labden-15-oic acid; D3: 6-oxocativic acid.

**Table 2 plants-09-00593-t002:** Effect of leaf litter on the germination rate (*GR*) and time to 50% of germination (*T*_50_) of the analysed species. N = 4.

	Treatment	*R. sphaerocarpa*	*C. multiflorus*	*L. stoechas*	*C. crispus*	*C. salviifolius*
	C	9.8 ± 1.8 a	17.4 ± 3.8 a	15.6 ± 1.4 a	27.4 ± 1.9 a	32.2 ± 2.5 a
	WL ⅓	9.8 ± 0.5 a	21.7 ± 4.0 a	27.1 ± 5.4 b	31.5 ± 2.2 ba	29.6 ± 2.8 a
***GR***	WL ⅔	11.6 ± 2.7 a	20.2 ± 2.3 a	24.9 ± 2.6 b	26.9 ± 1.9 a	33.0 ± 3.4 a
	UL ⅓	9.7 ± 1.4 a	19.5 ± 2.4 a	31.2 ± 1.3 b	45.3 ± 14.0 b	32.85 ± 0.4 a
	UL ⅔	13.8 ± 4.7 b	17.1 ± 4.0 a	28.5 ± 3.4 b	39.7 ± 8.8 b	35.8 ± 3.4 a
	C	8.33 ±0.7 a	16.8 ± 3.0 a	13.1 ± 1.5 a	23.9 ± 1.0 a	31.7 ± 5.9 a
	WL ⅓	8.50 ± 0.4 a	21.6 ± 3.7 a	22.5 ± 10.4 ab	23.3 ± 2.9 a	28.8 ± 3.8 a
***T*_50_**	WL ⅔	9.18 ± 1.3 a	19.3 ± 3.7 a	24.0 ± 2.2 b	23.8 ± 0.7 a	32.7 ± 3.7 a
	UL ⅓	8.15 ± 0.3 a	19.3 ± 8.2 a	27.1 ± 0.9 b	31.6 ± 3.6 b	31.7 ± 0.9 a
	UL ⅔	10.2 ± 4.3 a	16.9 ± 4.8 a	29.4 ± 3.9 b	28.2 ± 3.2 b	35.6 ± 4.0 a

*GR*: days required for germination; *T*_50_: days required for 50% of germination; C: vermiculite control; WL: washed leaf litter; UL: untreated leaf litter; 1/3: 1.33 dm^3^ of vermiculite and 0.66 dm^3^ of leaf litter; 2/3: 0.66 dm^3^ of vermiculite and 1.33 dm^3^ of leaf litter. a, b: same letter means no significant difference between treatments (*p* > 0.05).

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
