# Peer review of "Effect of Leaf Litter from Cistus ladanifer L. on the Germination and Growth of Accompanying Shrubland Species"

_plants, 2020, doi:10.3390/plants9050593_

Round 1
Reviewer 1 Report
In my opinion the manuscript “Effect of leaf litter from Cistus ladanifer L. on the germination and growth of accompanying shrubland species” is well written and presents enough new data. The research provides important information about the involvement of leaf litter in the allelopathic activity of C. ladanifer in Mediterranean ecosystems, and generated data are significant to the programmes of regeneration and development of the Mediterranean shrublands. The results are sufficiently clear described and well documented. This manuscript, in my opinion, is acceptable for publication after minor revision. There really are only a few comments that could be addressed by the authors:
Table 2, second column. Please, change LW into WL.
Author Response
Reviewer 1:
“Table 2, second column. Please, change LW into WL”
-It has changed
Reviewer 2 Report
The review of the manuscript draft by Gallego et al. submitted to Plants MDPI.
The paper is quite nicely written and is focused on interesting topics of allelopathy. However as the authors have stated the allelopathic effect of Cistus ladanifer is well documented in scientific literature. The authors should more clearly stated the novelty of the present study. Also this study seems to be a little out of date. The authors cited 78 papers, but only 5 of them, which consist on 6.4% were published in the last 5 years. There are also too many papers published in Spanish.
The other serious issue concern the sentences in lines 112-115. You have admitted that you used for germination tests soil taken from natural habitat and that seeds of Cistus ladanifer had been already in the soil, what was manifested by GP, which exceeded 100%. In such situation I am extremely caution and very skeptical to all of the results. In my opinion different amount of all of tested seeds could be already present in the soil.
When you used an abbreviation for experimental variants such as L and WL you should used them instead full name throughout whole manuscript. In L118 you have used a term unwashed leaf litter – is it the same as L, or some new variant?
For me the parameter GRI is not informative, as it will be the same for fast and slow germinating species with the same germination rate. From my point of view germination rate, and such parameters as T1, T50 and Tmax would be more informative about dynamic of the germination process and disturbance in it.
The word potentiate in L187 does not suit there.
In such cases like L236 and L248 (L270) you should introduced also name of the first author of cited references.
M&M section
You should mention all of used experimental variants. There is lack of L and WL (L322-326).
In L377 you referred to procedure according to Corral et al. 1990, however the authors studied only 2 from 3 listed by you species and recommended other procedure as the best one to stimulate seed germination (30 min incubation at 100 °C).
The word substrates in L343 does not suit there, as well as specimen in L368.
Author Response
Reviewer 2:
1.¨The authors have stated the allelopathic effect of Cistus ladanifer is well documented in scientific literature. The authors should more clearly stated the novelty of the present study¨
It is true that the allelopathic activity of C.ladanifer has been already proved, as it is manifested in the referred work. Nevertheless, the purpose of this work is not the demonstration of this species’ allelopathic potential but to go forward. On the one hand, the aim is to highlight the importance of leaf litter in an ecosystem functioning by influencing the composition of the species that make up that ecosystem. Most studies show the positive contribution of leaf litter, but our study also aims to emphasize in its possible negative implications. Our study goes a little further demonstrating the compounds toxicity with allelopathic activity by showing that allelopathy is an interaction that must be considered to understand the structure and functioning of an ecosystem.
In the introduction, a paragraph has been added in order to strengthen this idea.
2.“Also this study seems to be a little out of date. The authors cited 78 papers, but only 5 of them, which consist on 6.4% were published in the last 5 years. There are also too many papers published in Spanish.”
Some references have been removed and replaced for more current ones. Some references in Spanish has been removed as well.
3.“The other serious issue concern the sentences in lines 112-115. You have admitted that you used for germination tests soil taken from natural habitat and that seeds of Cistus ladanifer had been already in the soil, what was manifested by GP, which exceeded 100%. In such situation I am extremely caution and very skeptical to all of the results. In my opinion different amount of all of tested seeds could be already present in the soil.”
The leaf litter was collected from the habitat where C. ladanifer grows, under the C. ladanifer individuals. This leaf litter could contain capsule remains (where C. ladanifer seeds are found) that were impossible to separate from the leaves and, therefore, those seed germinated during the tests. There could not be seeds of the rest of the species owing to the fact that dominance of C. ladanifer in that community was over 80% and there were no individuals of other species around.
Despite this, when the major germination of C. ladanifer compared to the seeded ones was noticed, an experience was prepared with a lead litter tray under the same conditions as the rest of trays. The result was some C. ladanifer seedling germinated.
- “When you used an abbreviation for experimental variants such as L and WL you should used them instead full name throughout whole manuscript. In L118 you have used a term unwashed leaf litter – is it the same as L, or some new variant?”
Unwashed leaf litter is the same as L. The text terms have been unified.
- “For me the parameter GRI is not informative, as it will be the same for fast and slow germinating species with the same germination rate. From my point of view germination rate, and such parameters as T1, T50 and Tmax would be more informative about dynamic of the germination process and disturbance in it.”
GRI has been eliminated and replaces by T50 parameter.
6.“The word potentiate in L187 does not suit there”
Has been changed to “increases”
- In such cases like L236 and L248 (L270) you should introduced also name of the first author of cited references. OK
The name of the first author has been added.
8.You should mention all of used experimental variants. There is lack of L and WL (L322-326).
These terms have been completed and restated: WL: washed leaf litter; UL: untreated leaf litter
- In L377 you referred to procedure according to Corral et al. 1990, however the authors studied only 2 from 3 listed by you species and recommended other procedure as the best one to stimulate seed germination (30 min incubation at 100 °C).
This reference has been replaced.
10.The word substrates in L343 does not suit there, as well as specimen in L368.
Has been changed
Reviewer 3 Report
1) The title and the abstract look reasonable. The Reviewer suggests for abstract to add separately the description of effects on seed germination and then on seedling growth + to add the effects on the Cistus ladanifer itself.
2) On the other hand, C. multiflorus and C. salvifolius were only negatively affected in
25 seedling growth.
Better: “at the stage of seedling growth” or a similar phrase. Pls, check language for all the text.
3) Table 1. Units could not mg/mg dw, moreover not even mg/g since the Authors have values: K-3,7 2564.7±114.7
4) Table 1. It is better to provide % in the same form as means +- SD for the 3 independent experiments, not just the divided averages from the table.
5) Figure 1. Figure legend. Leaf litter from which plant?
6) Figure 1. The quality of letters and names for the figure is not good.
6) Table 2 and the description. What was the total number of seeds, it’s better to express the results in % or in numbers/total number.
7) Figure 2. The same problems with clear numbers, they are not readable.
8) Figure 2.
- a) Root size, what is it? Root length?
- b) Age of the roots, was is the same for all the treatments?
- c) How is it possible to compare root length when the germination time was different?
9) Figure 1. Why the % of germination was different for the species?
10) Figure 2. It doesn’t make sense to simple comparisons of vermiculate with leaf litter since the mineral components/nutrients/ions etc. are quite different for the two substrates.
11) Figure 2. C. ladanifer was the most affected plant, is it correct? What’s about allelopathy then?
12) Figure 3. C. ladanifer is the most affected again, correct?
13) Figure 3. Why the error bars are not centered relative to the bars?
14) Figure 4. It doesn’t make sense to average the results for all the species.
15) Where are the germination results for C. ladanifer?
16) Figure 4. Where is the y scale for root length? How could % could be the same as mm?
17) Figure 5. Same as for figure 4.
18) It has been shown that leaf litter with high concentrations of phenolic compounds hinder the
84 growth seedlings [42,43]. Growth of seedlings. Language problems. Check the text, pls.
19) Methods. 354 - Germination rate (GR): The germination rate is an arithmetic mean that indicates the days
355 required for germination [76]. It was calculated using the formula cited by [77]:
Formula mentions G, not GR.
20) - Germination rate index (GRI): it is defined as the average number of seeds that germinate every
361 day. GRI was calculated using the formula of [78]:
Not a very useful parameter. Was it even germination with the same number of germinated seeds every day?
21) germination tests were conducted in a greenhouse under the following conditions: 12 hours of
347 illumination (20-25ºC) and What was the illumination intensity? It’s important for so huge seedlings.
22) of the flavonoids and diterpenes present in the exudate of C. ladanifer
382 [39, 40].Why were they in exudates and not in the decomposed leaves?
23) Would be good to add an initial figure with the growing plants and arrows showing the collected samples of litter.
24) Maguire, J.D. Speed of germination in selection and evaluation for seddling emergence and vigor. Crop Sci.
568 1962, 2, 176-177.Misprint.
25) Any direct effects of the extracted compounds on the measured parameters known?
26) Basically here are the reasonable results which are not opposite to the hypothesis that some effect of leaf litter may exist. How do the Authors explain the negative effects on C. ladanifer itself? More points to discuss.
27) Any correlations between the measured parameters and distribution of the mentioned species in the plant communities?
28) Figure 1. What’s about the germination parameters for C. ladanifer itself?
29) Plenty to reconsider, change/add.
Author Response
Reviewer 3
1.“The Reviewer suggests for abstract to add separately the description of effects on seed germination and then on seedling growth + to add the effects on the Cistus ladanifer itself”.
Has been changed
- On the other hand, C. multiflorus and C. salvifolius were only negatively affected in seedling growth. Better: “at the stage of seedling growth” or a similar phrase. Pls, check language for all the text.
It has been changed to “at the stage of seedling growth”
- Table 1. Units could not mg/mg dw, moreover not even mg/g since the Authors have values: K-3,7 2564.7±114.7
It is a mistake. The values have been corrected to mg/g dw
- Table 1. It is better to provide % in the same form as means +- SD for the 3 independent experiments, not just the divided averages from the table.
It has been changed.
- Figure 1. Figure legend. Leaf litter from which plant?
Leaf litter of Cistus ladanifer
- Figure 1. The quality of letters and names for the figure is not good.
The figures have been modified so as to enhance their visibility
- Table 2 and the description. What was the total number of seeds, it’s better to express the results in % or in numbers/total number.
In this table, the GR has been expressed: The germination rate is an arithmetic mean that indicates the days required for germination and the T50: The time (days) to reach 50% germination.
- Figure 2. The same problems with clear numbers, they are not readable.
The figures have been modified so as to enhance their visibility
- Figure 2.
- a) Root size, what is it? Root length?
- b) Age of the roots, was is the same for all the treatments?
- c) How is it possible to compare root length when the germination time was different?
It has been changed to root length .
As shown in a lot of allelopathy work, root size is one of the most sensitive parameters to the compounds toxicity with allelopathic activity. Although it can be thought the smaller size of the root may be due to the fact that the seeds were born later, the experiences were maintained for long enough time (50 days) to reach a larger size. This can be verified with Cytisus multiflorus: the germination was not affected, neither was the speed of germination but the size of the roots was.
- Figure 1. Why the % of germination was different for the species?
Each species has specific characteristics and the final percentage reached by each one is different. This disparity is noticeable at controls and treatments.
- Figure 2. It doesn’t make sense to simple comparisons of vermiculate with leaf litter since the mineral components/nutrients/ions etc. are quite different for the two substrates.
Vermiculite is used as control. This is a inert substance and this allows us to quantify any effect derived from leaf litter.
- Figure 2. C. ladanifer was the most affected plant, is it correct? What’s about allelopathy then? Figure 3. C. ladanifer is the most affected again, correct?
- ladanifer is equally affected, significantly, just like the other species.
When allelochemicals that are released from a plant affect growth or germination of individuals of the same species, the term allelopathy is replaced by the term autotoxicity. Autotoxicity is allelopathy.
- Figure 3. Why the error bars are not centered relative to the bars?
There was a mistake in the deviation bars, and it has been rectified.
- Figure 4. It doesn’t make sense to average the results for all the species. Figure 4. Where is the y scale for root length? How could % could be the same as mm?. Figure 5. Same as for figure 4.
Figures 4 and 5 are intended to visualize, in a general and clearer way, the effect of the leaf litter on a community final composition. These figures can serve as a summary and general ideal, so they have been put in this discussion. We consider they are interesting as a clearer way to visualize a general conclusion, as well.
The figures have been changed for a better view of the axes.
- Where are the germination results for C. ladanifer?
This is explained in line results: L130-133
- It has been shown that leaf litter with high concentrations of phenolic compounds hinder the growth seedlings [42,43]. Growth of seedlings. Language problems. Check the text, pls.
It has been changed
- Methods. 354 - Germination rate (GR): The germination rate is an arithmetic mean that indicates the days required for germination [76]. It was calculated using the formula cited by [77] Formula mentions G, not GR.
It has been changed
- Germination rate index (GRI): it is defined as the average number of seeds that germinate every day. GRI was calculated using the formula of [78]: Not a very useful parameter. Was it even germination with the same number of germinated seeds every day?
It has been replaced by the T50
- of the flavonoids and diterpenes present in the exudate of C. ladanifer [39, 40]. Why were they in exudates and not in the decomposed leaves?
These compounds are also found in the leaf litter, in smaller quantities compared to the fresh leaves’ exudate. Table 1 shows the amount of these compounds in the leaf litter.
- Would be good to add an initial figure with the growing plants and arrows showing the collected samples of litter.
Providing these images is a good idea, but there is a problem: as you surely know, Spain is currently in a sanitary state alarm and we are teleworking at home. Unfortunately, I cannot currently accede to the fotographs that show these experiences as they are stored on a computer located at the University.
- Maguire, J.D. Speed of germination in selection and evaluation for seddling emergence and vigor. Crop Sci. 1962, 2, 176-177. OK
This reference has been eliminated and substituted by the correct reference for the T50 calculation
- Any direct effects of the extracted compounds on the measured parameters known?
In the referred work in the text, which I present below, the involvement of these compounds in the allelopathic activity of C.ladanifer is revealed:
- Chaves, N.; Alías, J.C.; Sosa, T. Phytotoxicity of Cistus ladanifer : Role of allelopathy. Allelopath. J. 2016, 38, 113-131.
- Chaves, N.; Escudero, J.C. Allelopathic effect of Cistus ladanifer on seed germination. Ecol. 1997, 11: 432-440.
- Dias, A.S.; Dias, L.S.; Pereira, I.P. Activity of water extracts of Cistus ladanifer and Lavandula stoechas in soil on germination and early growth of wheat and Phalaris minor. J. 2004, 14, 59–64.
- Alías, J.C. Influence of Climatic Factors on the Synthesis and Activity of Phytotoxic Compounds Secretred by Cistus ladanifer Ph.D. Thesis, Universidad of Extremadura, Extremadura, Spain, 2006.
- Chaves, N.; Sosa, T.; Escudero, J.C. Plant growth inhibiting flavonoids in exudate of Cistus ladanifer and in associated soils. Chem. Ecol. 2001, 27: 623-631. Chaves, N.; Sosa, T.; Alías, J.C.; Escudero, J.C. Germination inhibition of herbs in Cistus ladanifer L. soil: Possible involvemente of allelochemicals. Allelopath J. 2003, 11, 31-42.
- Basically here are the reasonable results which are not opposite to the hypothesis that some effect of leaf litter may exist. How do the Authors explain the negative effects on C. ladanifer itself? More points to discuss.
In the papers “-Alías, J.C.; Sosa, T.; Escudero, J.C.; Chaves, N. Autotoxicity against germination and seedling emergence in Cistus ladanifer L. Plant Soil 2006, 282, 327-332. and -Chaves, N.; Ferrer, I.; Alías, J.C. Autotoxicity of diterpenes present in leaves of Cistus ladanifer L. Plants 2019” ” it is argued the autotoxicity of C. ladanifer.
In the discussion, a paragraph has been added in order to expose the possible ecological implication of autotoxicity in C.ladanifer and in other species where this interaction has also been detected.
- Any correlations between the measured parameters and distribution of the mentioned species in the plant communities?
The negative effect of C.ladanifer’s leaf litter in the germination and the seedling growth of the studied species could explain the low diversity and richness of the species in communities dominated by C.ladanifer, as it is exposed in the references:
- Núñez-Olivera, E.; Martinez-Abaigar, J.; Escudero, J.C.; García-Novo, F. A comparative study of Cistus ladanifer shrublands in Extremadura (CW Spain) on the basis of woody species composition and cover. Vegetatio 1995, 117, 123-132.
- Herranz, J. M.; Ferrandis, P.; Copete, M. A.; Duro, E. M.; Zalacaín, A. Effect of allelopathic compounds produced by Cistus ladanifer on germination of 20 Mediterranean taxa. Plant Ecol. 2006, 184, 259-272.
- Tárrega, R.; Luis, E.; Valbuena, L. Eleven years of recovery dynamic after experimental burning and cutting in two Cistus Acta Oecol. 2001, 22, 277-283.
Calvo, L.; Tárrega, R.; Luis, E.; Valbuena, L.; Marcos, E. Recovery after experimental cutting and burning in three shrub communities with different dominant species. Plant Ecol. 2005, 180, 175-185.
- Figure 1. What’s about the germination parameters for C. ladanifer itself?
This is explained in line results: L130-133
Round 2
Reviewer 2 Report
The authors have provided proper modification into manuscript and send reasonable answers to all my questions in the correspondence. In my opinion all the modification increased the value of this paper, however I am still not sure if such high cited journal is right place to publish this research. This decision belongs to the Editor, if yes I am for publishing this paper in the present form.
Author Response
I send you attached the paper with the changes proposed by reviewer 3, where figures 2 and 6 have been added.
In figure 6 shows an image of the trays in the greenhouse with the different treatments and figures 2 shows the germination in the control and UL â…” treatment of two of the species with which it has been tested: R. sphaerocarpa (where there is no effect on germination) and L. stoechas (one the most affected species).
Reviewer 3 Report
- Would be good to add an initial figure with the growing plants and arrows showing the collected samples of litter.
Providing these images is a good idea, but there is a problem: as you surely know, Spain is currently in a sanitary state alarm and we are teleworking at home. Unfortunately, I cannot currently accede to the fotographs that show these experiences as they are stored on a computer located at the University.
Author Response
I send you attached the paper with the changes proposed, where figures 2 and 6 have been added.
In figure 6 shows an image of the trays in the greenhouse with the different treatments and figures 2 shows the germination in the control and UL â…” treatment of two of the species with which it has been tested: R. sphaerocarpa (where there is no effect on germination) and L. stoechas (one the most affected species).
Round 3
Reviewer 3 Report
Basically the paper fits the volume and the subject of the journal.
The quality of the research with reasonable level of resolution is also satisfactory for the journal.
Figure 7 is not required or better to be combined with figure 2. + a scale for the size.
Table 1. Still mg/mg in bold are not corrected.
Figure 2. Just letters are OK + a scale in nanometers or in nautical miles. In cm is OK.
Figure 5A. Errors bars are required.
Figure 5B. Root length should be in % of control for each species + the error bars or SDs.
Same for parts of figure 6 (6a and 6B) as for parts of figure 5 (5A and 5B).
Author Response
Reviewer 3:
Figure 7 is not required or better to be combined with figure 2. + a scale for the size.
Figure 7 has been eliminated.
Table 1. Still mg/mg in bold are not corrected.
It has been changed
Figure 2. Just letters are OK + a scale in nanometers or in nautical miles. In cm is OK.
Scale (cm) has been added
Figure 5A. Errors bars are required.
Figure 5B. Root length should be in % of control for each species + the error bars or SDs.
Same for parts of figure 6 (6a and 6B) as for parts of figure 5 (5A and 5B).
Figures 5 and 6 has been changed
Round 4
Reviewer 3 Report
The quality of presentation is low, not all the recommendations are realised. The next stage is to reject the MS. The best is to resubmit the MS later.
Author Response
The paper has been read carefully and the errors detected have been corrected.